# *Vive la Difference*! The Effects of Natural and Conventional Wines on Blood Alcohol Concentrations: A Randomized, Triple-Blind, Controlled Study

**DOI:** 10.3390/nu11050986

**Published:** 2019-04-30

**Authors:** Federico Francesco Ferrero, Maurizio Fadda, Luca De Carli, Marco Barbetta, Rajandrea Sethi, Andrea Pezzana

**Affiliations:** 1FFF IMAGE srls, 10143 Torino, Italy; 2Clinical Nutrition Unit, Città della Salute e della Scienza, 10126 Torino, Italy; mfadda@cittadellasalute.to.it; 3Clinical Nutrition Unit, ASL Città di Torino, 10128 Torino, Italy; l.decarli@unisg.it (L.D.C.); andrea.pezzana@unito.it (A.P.); 4MSquare Dynamics S.r.l., 35129 Padova, Italy; m.barbetta@msquaredynamics.com; 5Department of Environment, Land and Infrastructure Engineering (DIATI), Politecnico di Torino, 10129 Torino, Italy; rajandrea.sethi@polito.it

**Keywords:** alcohol, natural wine, blood alcohol content, breathalyzer, pesticides

## Abstract

Different alcoholic beverages can have different effects on blood alcohol concentration (BAC) and neurotoxicity, even when equalized for alcohol content by volume. Anecdotal evidence suggested that natural wine is metabolized differently from conventional wines. This triple-blind study compared the BAC of 55 healthy male subjects after consuming the equivalent of 2 units of alcohol of a natural or conventional wine over 3 min in two separate sessions, one week apart. BAC was measured using a professional breathalyzer every 20 min after consumption for 2 h. The BAC curves in response to the two wines diverged significantly at twenty minutes (interval T20) and forty minutes (interval T40), and also at their maximum concentrations (peaks), with the natural wine inducing a lower BAC than the conventional wine [T20 = 0.40 versus 0.46 (*p* < 0.0002); T40 = 0.49 versus 0.53 (*p* < 0.0015); peak = 0.52 versus 0.56 (*p* < 0.0002)]. These differences are likely related to the development of different amino acids and antioxidants in the two wines during their production. This may in turn affect the kinetics of alcohol absorption and metabolism. Other contributing factors could include pesticide residues, differences in dry extract content, and the use of indigenous or selected yeasts. The study shows that with the same quantity and conditions of intake, natural wine has lower pharmacokinetic and metabolic effects than conventional wine, which can be assumed due to the different agronomic and oenological practices with which they are produced. It can therefore be hypothesized that the consumption of natural wine may have a different impact on human health from that of conventional wine.

## 1. Introduction

In recent decades, wine consumption has been the subject of intense debate within the scientific community. On the one hand, wine has been linked to reduced risk for several chronic illnesses, such as cardiovascular diseases, osteoporosis, and diabetes [1]. On the other hand, international guidelines for cancer prevention emphasize the direct correlation between alcohol intake and cancer risk [2,3]. The positive health benefits provided by wine come primarily from compounds called polyphenols, which are natural antioxidants that help fight inflammation and improve plasma lipid profiles [4]. When consumed regularly and moderately, ethanol, the main alcohol component in wine, confers cardioprotective effects by acting directly on cardiomyocytes, blood circulation, and platelet aggregation [5]. Nonetheless, ethanol and its metabolite acetaldehyde are also responsible for adverse neurological, hepatic, and oncological consequences secondary to alcohol consumption [6,7]. Because of its potentially beneficial and harmful effects, many scientific organizations recommend that alcohol consumption be limited to lower-alcohol beverages, such as wine [8], and that such beverages be consumed moderately and responsibly, if at all [9,10].

In Italy, the Research Centre for Food and Nutrition of the Council for Agricultural Research and Economics (CREA-AN) has adopted guidelines issued by the National Institute for Research on Food and Nutrition (INRAN), which defines moderate alcohol consumption as an average daily allowance of no more than 2–3 units of alcohol for men, and 1–2 units for women. The standard value of a unit of alcohol in Italy is 12 g of ethanol [11].

As reported by the INRAN, there is a well-known linear correlation between blood alcohol concentration (BAC) and the deleterious effects of alcohol, particularly those involving the central nervous system [11]. The short-term neurotoxic effects of an elevated BAC include a state of euphoria or inebriation, slowed reflex and reaction times, diminished peripheral vision, and cognitive impairment [12].

The relationship between the amount of alcohol consumed and BAC is influenced by numerous factors, including the individual’s sex, age, body weight, liver volume and function, drinking habits, use of medications, medical conditions, and fasting or non-fasting state [13].

It has also been established that when equalized for alcohol content by volume, different beverages are absorbed at different rates and lead to different maximum concentrations (peaks) in BAC [14,15]. This study set out to determine whether the absorption of ethanol from two wines produced from the same grape (with similar alcohol and low sugar content) might be affected by differences in the farming and winemaking techniques used in their production. The approach was thus to compare the evolution of BAC of healthy male subjects after their consumption of 2 units (24 g of alcohol) of a natural wine (i.e., cultivated without pesticides and agrochemicals, fermented with wild yeasts, unfiltered, and with no fining) and after their consumption of the same amount of a conventional wine, one week apart and under the same experimental conditions.

## 2. Materials and Methods

Participants were administered samples of the same quantity of a natural wine (NW) and a conventional wine (CW) one week apart, and their BAC was measured at regular intervals for two hours after their oral ingestion of the wine.

### 2.1. Selection of the Natural and Conventional Wines

In the absence of clear national or international legislation on the definition of natural wine, it was decided for the purpose of this study to compare the effects of consuming two near-identical wines differing only in the farming management and vinification protocols adopted in their production. Over 300 wines were purchased for testing by an independent laboratory specializing in alimentary analysis, in order to identify those suitable for comparison. Our intention was to identify the pair of wines, one natural and one conventional, with the best possible correspondence in grape variety, proximity of area of production, age of harvesting, alcoholic strength by volume, and low sugar content (<1.5 g/L). Two wines satisfied the inclusion criteria. Both were whites made from Cortese grapes grown in vineyards located within 10 km of each other in Piedmont, Italy. They were of the same vintage and aged in bottles for 12 months. Table 1 shows the main characteristics of the two wines selected for the trial. While they were similar in alcohol strength by volume and both were low in sugar content, they showed significant differences in volatile acidity, total dry extract, and sulfur dioxide concentrations, which can be attributed to the different farming and vinification processes used in their production.

As shown in the table, the two wines had the same percent of alcohol by volume, were both low in sugar content (<1.5 g/L), and were made from the same variety of grape, grown in the same geographic location. The grapes used for making the natural wine, however, were cultivated in the absence of pesticides or agrochemicals, other than those approved for organic farming by Council Regulation (EC) No 834/2007 [16]. The natural wine was fermented without the use of selected yeasts or fining agents. It was also left unfiltered, and no sulfites were added. In contrast, the grapes used for making the conventional wine were grown using regulated synthetic pesticides and agrochemicals and fermented with selected yeasts; in addition, the entire winemaking process was based on conventional methods permitted by Italian law, including filtration and the addition of sulfur dioxide. The wines were subjected to additional tests for pesticides, revealing the presence of trace concentrations of iprovalicarb (45 µg/kg) and fenhexamid (120 µg/kg) in the conventional wine.

### 2.2. Study Design

The study was a randomized, triple-blind, controlled trial. Each phase of the trial was conducted as a triple-blind test, as the type of wine administered was unknown to (a) the research participants, (b) the individuals who administered the wines, and (c) the individuals who assessed the outcomes.

Three teams worked on the experiment. The first team designed the study, selected the volunteers, and set up the samples for testing. All the labels on the bottles of wines used for the experiment were masked, and the wines were identified by a four-digit code. The first two digits indicated the day on which the test was performed, while the second two indicated the bottle index randomly assigned to each bottle. The second team administered the doses of wine to the subjects and recorded the resulting data, and the third team processed the data, cross-referencing the matrix of wine types received from the first team with the BAC data of the individual participants collected by the second team. The second team was not aware of the type of the wine being administered to the participants, and the third team never met the participants. 

The study was conducted as a crossover trial: (i) it involved a single study group, in which each subject received two treatments in turn; (ii) each subject served as his own control, and the comparison of treatments was made within the subjects; (iii) two or more treatments were administered in a randomized order; (iv) the subjects were healthy; and (v) there was a sufficient period of washout between treatments, in order to ensure that the treatments would not interfere with each other and to avoid carry-over effects. The participants intended to receive one type of wine or the other during the first week’s trial (and thus the other wine the following week) were chosen randomly. Among the 167 eligible subjects, 55 subjects were drawn by a random number generator. In addition, at the start of the test, the matrices containing the match between the subjects and the bottles were generated by a random number generator. The researchers in the first team, who were in charge of selecting the subjects, did not know which wine the subjects would be assigned to, because the matrix with the assignments was in the hands of the second team. The subjects were also unaware of the type of wine they would drink, having been informed that they would be drinking a sample of wine both weeks, without specifying the type of wine or whether it would be the same or different each week.

### 2.3. Subjects

All participants in the study were university student volunteers, screened using the Italian Ministry of Health Surveillance Questionnaire (known as PASSI) to collect data on their height, weight, body mass index (BMI), dietary habits, and use of prescription medicines. The questionnaire PASSI, validated and promoted by The Italian Ministry of Health, The National High Institute of Public Health, and The National Centre for Disease Prevention and Control for the surveillance of behavioral risk factors, as well as for the monitoring of chronic disease and prevention programs, is commonly accepted as a basis for national and regional reports on alcohol consumption in Italy, and is unanimously recognized as valid for extrapolating behavioral indications and health recommendations for the general population. It also contains a special section in which the subject declares, under his own responsibility, the validity of the data provided. The inclusion criteria were that subjects be healthy Caucasian male university students aged 18 to 30 with a BMI between 18.5 kg/m^2^ and 25 kg/m^2^ and who were undergoing no drug therapy, consumed on average 4 alcohol units per week, and were able to understand the purpose of the study and thus provide written informed consent. Exclusion criteria included the use of prescription medicine for chronic conditions, any pathology that might interfere with alcohol metabolism, or habitual consumption of more than 4 units of alcohol per week. The 55 males randomly selected out of the 167 eligible subjects recruited were of median age 23 years (r 21–24), median weight 69 kg (r 65–78), median height 178 cm (r 174–183), and median BMI 22 kg/m^2^ (r 20.8–22.9).

Ethical approval was provided by the Polytechnic University of Turin (Department Resolution No 1037/2018, 01-30-2018) in compliance with the Helsinki Declaration. Recruitment was limited to male subjects, as the study required consumption of 2 units of alcohol, which exceeds the maximum recommended daily amount for women (INRAN, 2003). At the end of the study, an information and awareness campaign was carried out to promote alcohol awareness and responsible use among the entire student body at the Polytechnic University of Turin. 

### 2.4. Administering the Two Wines

The test took place over two sessions held one week apart. At the first session, each subject was given 3 min to drink a single, unlabeled, and randomly selected 248 mL dose of either the natural wine or the conventional wine (the equivalent of 2 units, or 24 g of alcohol), distributed in a plain black wine tasting glasses. At the second session, seven days later, the subjects had to repeat the experiment, this time being administered the other type of wine to drink. 

After providing breath samples at the beginning of each session to verify a zero BAC, the subjects underwent a series of tests performed with a professional breathalyzer (“Alcotrue M”, Bluepoint MEDICAL GmbH & Co. KG, D-23923 Selmsdorf, Germany) to measure their blood alcohol levels at 20 min intervals for a total of 2 h after ingesting the sample (time intervals: T0, T20, T40, T60 T80, T100, T120). 

The subjects were required to abstain from drinking alcohol for seven days, from smoking for 8 h, and from eating for at least 4 h prior to both sessions, in order to minimize possible interference from alcohol, smoking, or food during the previous week and the previous hours. The unlabeled wines were served at 21 ± 1 °C, and the same temperature was maintained in the testing room throughout the test to prevent any temperature-dependent interference on BAC [17]. The room had artificial lighting, and the two tasting sessions took place at the same time of day to minimize any possible interference. 

### 2.5. Pharmacokinetic Analysis 

A professional breathalyzer was used to estimate the following pharmacokinetic parameters: BACs, expressed in g/L at time intervals T0, T20, T40, T60, T80, T100, and T120. 

### 2.6. Statistical Analysis

The sample size was calculated on the basis of the main expected outcome, defined as the difference between BAC after drinking a fixed dose of natural wine, and BAC after drinking the same dose of conventional wine. Using data in the literature on subjects similar to those participating in our study, it was calculated that for an effect size of 0.67 and a two-tailed alpha error of 0.05, 50 subjects would be needed to obtain 90% power. As a precautionary measure, the sample size was set at 55 subjects. Continuous variables were expressed as medians and interquartile ranges (IQR), and categorical variables as percentages and absolute frequencies. The Student’s *t*-test for paired samples was used to detect differences in BAC at each of the time intervals (T0, T20, T40, T60 T80, T100, T120), and to detect differences in AUC as well. The resulting data was graphically represented using box-and-whisker plots. The level of significance was set at *p* ≤ 0.05. All statistical analysis was performed with the MedCalc Statistical Software version 19 (MedCalc Software bvba, Ostend, Belgium; https://www.medcalc.org; 2019).

## 3. Results

Breathalyzer measurements obtained at regular 20 min intervals were used to plot concentration–time curves of each subject’s BAC response to the natural and conventional wines. These can be seen in Appendix A, Figure A1, Figure A2, Figure A3, Figure A4 and Figure A5. Superimposition of the pairs of curves reveals that each subject had his own distinct pattern of alcohol pharmacokinetics, as is evident, for example, for subjects 21, 25, 43, and 44, randomly extracted from the sample (Figure 1).

By summing all of the subjects’ BAC values at the different time intervals and dividing by the number of subjects, the average BAC curves were calculated. Figure 2 shows the differences in the average BAC levels registered after the ingestion of the natural versus the conventional wine. 

It can be seen that the rate of increase in BAC in response to the two wines diverges significantly at the T20 mark, with natural wine inducing lower levels than conventional wine, at 0.40 versus 0.46 (*p* < 0.0002) (Figure 3); the average BAC of the natural wine is also significantly lower at T40, 0.49 versus 0.53 (*p* < 0.0015) (Figure 4).

The BAC peaks occur between T40 and T60 for both wines. The difference in values of the maximum BAC levels after the ingestion of the natural wine (NW) is significantly lower than after the ingestion of conventional wine (CW), at 0.52 versus 0.56, respectively (*p* < 0.0002) (Figure 5). The curves continue to approach each other until the T80 mark, and then they intersect. After this point, the conventional wine is associated with a slightly lower BAC; the curves gradually converge and largely overlap toward the end.

The AUC was calculated from T0 to T120 using the trapezoidal method. This parameter proved not to be significant (*p* = 0.13). This means that although the increasing BAC in response to the two wines differs at specific points along the curve, the overall variation does not reach significance when the curve is considered as a whole (Figure 6). 

It is interesting to note that, with the exception of the peak, the median BAC measured in response to the natural wine is consistently below 0.5 g/L, the maximum legal drink driving limit in many countries. In contrast, the BAC in response to conventional wine not only exceeds the legal driving limit at its maximum peak, but approaches the limit at T20 and exceeds it at T40.

## 4. Discussion

Our findings indicate that the peak BAC reached after drinking a natural wine is significantly lower than after drinking the same amount of a conventional wine with a similar total alcohol strength by volume. The alcohol in natural wine is absorbed more slowly than that in conventional wine, as can be seen by the discrepancy between the BAC measurements at T20 and at T40 (Figure 2). Ethanol is absorbed into the bloodstream mainly through the jejunum via passive diffusion, and down the concentration gradient between the small intestine and the capillaries [13]. Numerous factors can influence the absorption rate of alcohol: the type of beverage and manner of ingestion (total alcohol content, the concentration of alcohol, whether or not it is consumed as a single dose or as multiple smaller doses), as well as the intrinsic characteristics of the subject (mucosal integrity of the intestine, efficient blood flow, the presence or absence of food in the stomach, and alcohol dehydrogenase activity in the gastric mucosa) [13]. This study was designed to rule out possible causes for differences related to the mode of consumption or the intrinsic characteristics of the subjects. The causes can thus be attributed to differences in the non-alcoholic component of the two wines. 

Chemico-physical analysis of the samples revealed substantial differences in the total dry extract of the two wines. This is a direct consequence of differences in farming and winemaking practices, with the absence of filtration processes in natural wine likely to be a key factor. The total dry extract of a wine contains all of its non-volatile substances, such as sugars, polyphenols, fibers, and minerals. The total dry extract contained in the dose of natural wine was 1.67 g higher than in the conventional wine used for testing. This may affect gastric emptying time, and consequently, the absorption rate of ethanol [18].

The total sulfur dioxide content in the two wines also differed, with conventional wine containing the larger share. Sulfur dioxide has antioxidant and antiseptic properties that inhibit the growth of certain strains of yeast and bacteria during the various phases of winemaking [19]. Although the in vivo metabolic effects of sulfur dioxide have been widely studied [20], there have been no reports on its involvement in the absorption or metabolism of alcohol. 

Another important distinction between natural and conventional wines lies in the vinification process. Natural wine is the product of spontaneous fermentation by indigenous yeasts naturally found on the grapes, while conventional wines are produced using mixtures of laboratory-selected microorganisms. The presence of various strains of bacteria and yeasts during fermentation results in the development of different metabolites [21,22]. At present, the results of chemico-physical analysis of the samples used in this study are unable to provide precise information about these differences. Additional data may emerge thanks to the use of new technologies. In recent years, for example, high-field ^1^H nuclear magnetic resonance (^1^H-NMR) spectroscopy has allowed detailed investigation of wine metabonomics [23], and has demonstrated that the vinification protocol is one of the chief factors determining the amino acid, alcohol, and polyphenol make-up of two wines from the same geographic location [24]. Another study established that different production chains determine variations in the amount and type of antioxidants found in organic and biodynamic wines [25]. Comparable data on wines produced using the natural winemaking process are not yet available. 

Besides producing wines with different amino acid and polyphenolic profiles, differences in the natural and conventional fermentation pathways may also generate other molecules that interact with the absorption or with specific isoforms of alcohol dehydrogenase (ADH), the enzyme involved in breaking down alcohol. This would in turn lead to differences in the rates of metabolism of natural and conventional wines.

The polyphenolic content of wine has been found to alter the intestinal microbiota by stimulating the growth of bifidobacteria and lactobacilli and decreasing the numbers of clostridia and enterobacteria [26,27]. Prolonged alcohol abuse, on the other hand, can produce a state of intestinal dysbiosis, with overgrowth of proteobacteria [28]. It is unlikely, however, that these differences affect the absorption and metabolism of alcohol in the short term. 

It seems reasonable to expect the polyphenolic profile of two differently produced wines to have dissimilar effects on the individual’s microbiota. In any case, the wines used in this study were white, meaning they were not as rich in antioxidants as reds and rosés, so any variability due to the total content of antioxidants (particularly of resveratrol) was minimized [29].

A further possibility is that pesticide residues (Table 1) might interfere with the absorption, metabolism, and pharmacokinetics of alcohol in conventional wines, where contaminant analysis has revealed traces of the fungicides iprovalicarb and fenhexamid. Both are present within legal limits [30,31], and there have been no reports of acute intoxication or known effects on liver metabolism caused by their presence in wine [32,33]. However, understanding the toxicity of pesticides and their interaction with metabolic processes in vivo is extremely complex, given the vast number of simultaneously interacting molecules [34]. Therefore, it cannot be ruled out that synergistic interactions among the different contaminants might influence the absorption or metabolism of ethanol. 

The different kinetics observed for natural and conventional wines may have important clinical implications. Acute alcohol intoxication is one of the leading causes of emergency room visits [35], and approximately 5% of deaths from acute poisoning are attributable to alcohol [36]. Systemic toxic effects are proportional to BAC, and levels above 0.5 g/L are enough to impede normal daily activities. Concentrations above 4 g/L cause hypoventilation, which, if untreated, can lead to coma and death [37]. Our findings show that the peak BAC in response to natural wine is lower than that with conventional wine, meaning that natural wine is less likely to lead to alcohol intoxication.

Despite numerous mass media campaigns to promote responsible drinking, epidemiological evidence shows that they are largely ineffective in causing a reduction in alcohol consumption [38]. According to 2017 data from the Italian National Institute of Statistics (ISTAT), regular daily consumption of alcohol with meals is slightly on the decline, whereas occasional or irregular drinking outside of meals and binge drinking, particularly among youths below the age of 25, has increased dramatically [39].

It is estimated that 35% of road fatalities are linked to alcohol. Because of the dangers of alcohol-induced cognitive impairment, most European countries have passed laws making it an offense to drive with a BAC in excess of 0.5 g/L. The fact that in our study, among those subjects who drank 2 units of natural wine, only 56% exceeded the legal blood alcohol limit of 0.5 g/L, as opposed to 67% of those who drank the same amount of conventional wine, suggests that further investigations should be undertaken. 

A preliminary study by Bassani et al. (Marco Bassani, personal communication, 11 September 2018) compares the behavior of subjects who consumed natural wine or conventional wine prior to completing a simulated driving task. Subjects who drank conventional wine before the simulation tended to drive more aggressively than those who consumed an equal amount of natural wine. In particular, the natural wine drinkers drove consistently slower and committed fewer traffic violations than those who drank conventional wine.

## 5. Conclusions

To our knowledge, this is the first scientific study to compare the pharmacokinetics of alcohol of a conventional wine with that of a nearly identical natural wine. It analyzes the effects on BAC of drinking a natural wine or of an equal amount of conventional wine, both from the same production area and variety, with a similar alcohol content and low residual sugar. The BAC level 20 and 40 min after drinking the natural wine was lower than that after drinking the conventional wine, and the peak blood alcohol response to drinking natural wine was also lower than the peak response to drinking conventional wine. This supports the hypothesis that natural and conventional wines are metabolized differently.

The key strengths of this study are its randomized, triple-blind, controlled design, as well as its careful selection of the wines, both of which came from the same variety of grape and were virtually identical in many of their physical and chemical characteristics. Recruitment of a homogeneous group of subjects reduced the variability in individual kinetics and alcohol metabolism. A limitation of the study is the brief period (2 h) allotted to measuring the subjects’ BAC, and future trials may wish to extend the time used for testing. 

More work is needed to fully understand the relationship between natural wine and BAC. Currently, little data is available on other types of wine (red and rosé) and other segments of the population (women and the elderly). However, studies using higher doses of wine would pose ethical challenges related to exposing subjects to more alcohol than is considered safe. Additional studies using new technologies like ^1^H-NMR will make it easier to pinpoint differences in the chemical composition of natural and conventional wines. 

In the absence of specific laws and more precise laboratory data, the differences between natural and conventional wine must generally be imputed to differences in agricultural methods, winemaking processes, and preservation techniques, and their description is mainly relegated to expression of the consumers’ sensorial experience of the final product. The present study has been able to confirm that there are indeed objective differences in the absorption of natural wine and conventional wine.

Further research would be useful, with a view to developing a universal legislative framework for the regulation of natural wines, and for shedding light on differences in the public health implications of natural wine and conventional wine. Because it leads to a lower peak BAC than conventional wine, natural wine may be linked to a lower risk of alcohol intoxication. If future studies confirm this hypothesis, every effort should be made to include this information in public awareness and educational campaigns about responsible drinking. Given the growing international interest in natural wines [40,41], as well as consumers’ increasing demand for “natural” alcoholic beverages with a low environmental impact, further research should be undertaken to better understand the potential health benefits provided by natural wines.

## Figures and Tables

**Figure 1 nutrients-11-00986-f001:**
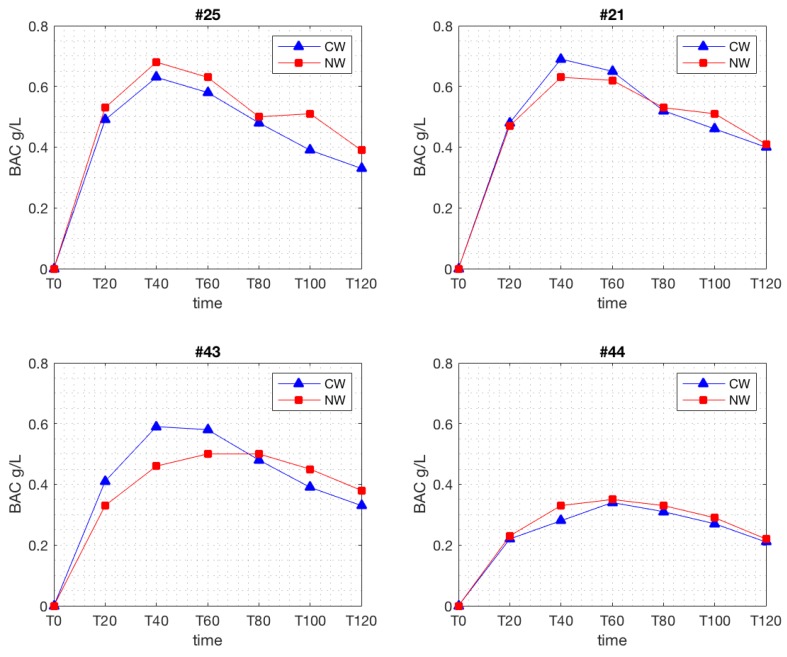
Randomly selected examples of blood alcohol concentration (BAC) levels measured every 20 min after the ingestion of the conventional wine (CW) and of the natural wine (NW) (g/L), in subjects #21, #25, #43, and #44.

**Figure 2 nutrients-11-00986-f002:**
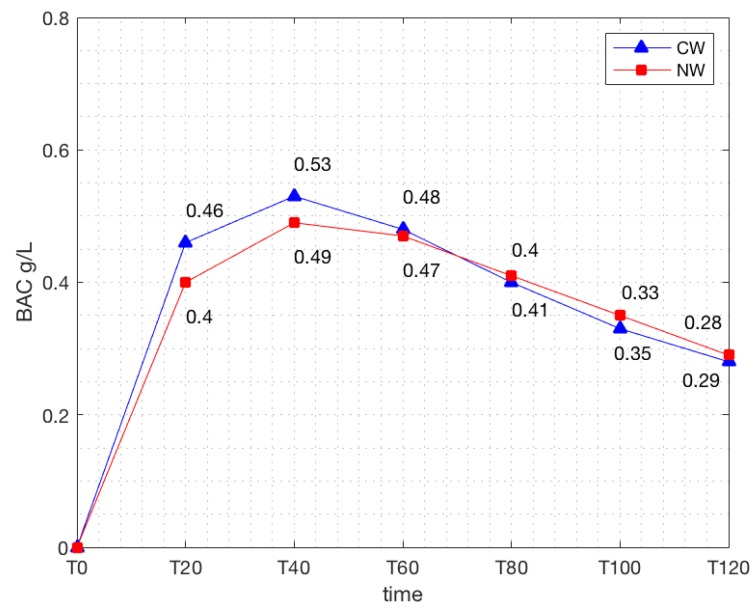
Comparison of the average blood alcohol concentration (BAC) levels measured every 20 min after the ingestion of the conventional wine (CW) and of the natural wine (NW) (g/L).

**Figure 3 nutrients-11-00986-f003:**
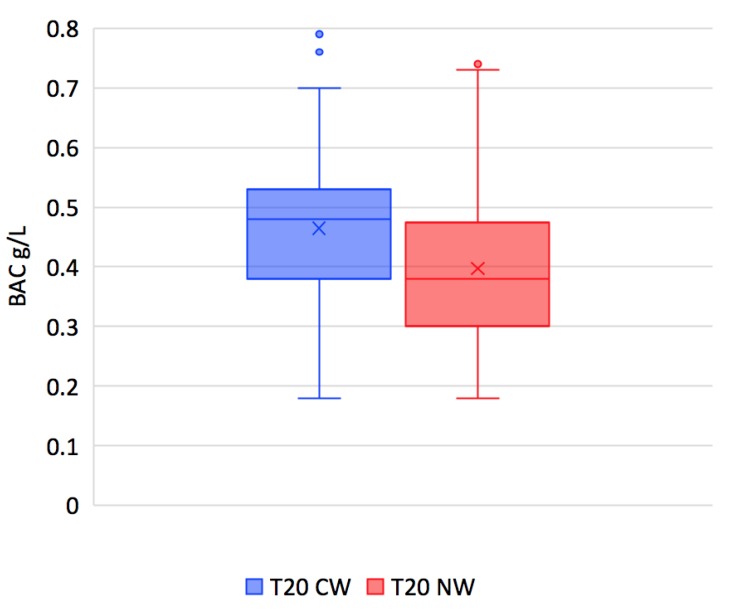
Box-and-whisker diagram of the blood alcohol concentration (BAC) levels at T20 after the ingestion of the conventional wine (CW) and of the natural wine (NW) (g/L) (*p* < 0.0002).

**Figure 4 nutrients-11-00986-f004:**
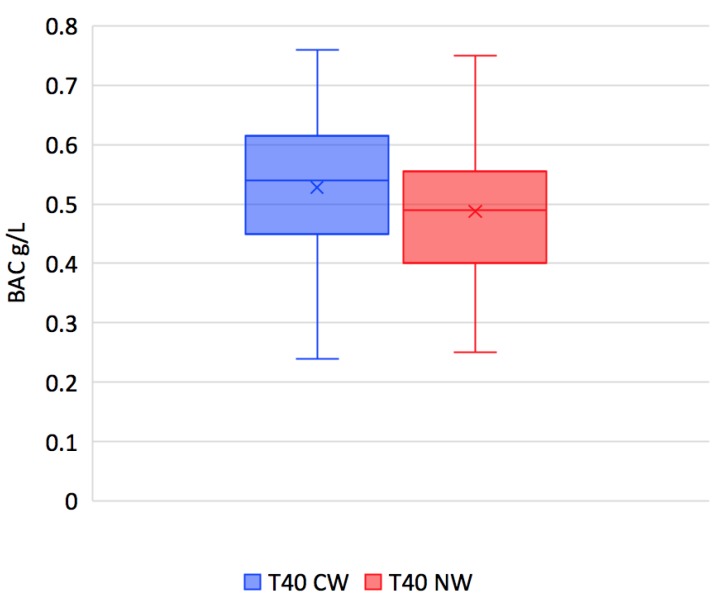
Box-and-whisker diagram of the blood alcohol concentration (BAC) levels at T20 after the ingestion of the conventional wine (CW) and of the natural wine (NW) (g/L) (*p* < 0.0015).

**Figure 5 nutrients-11-00986-f005:**
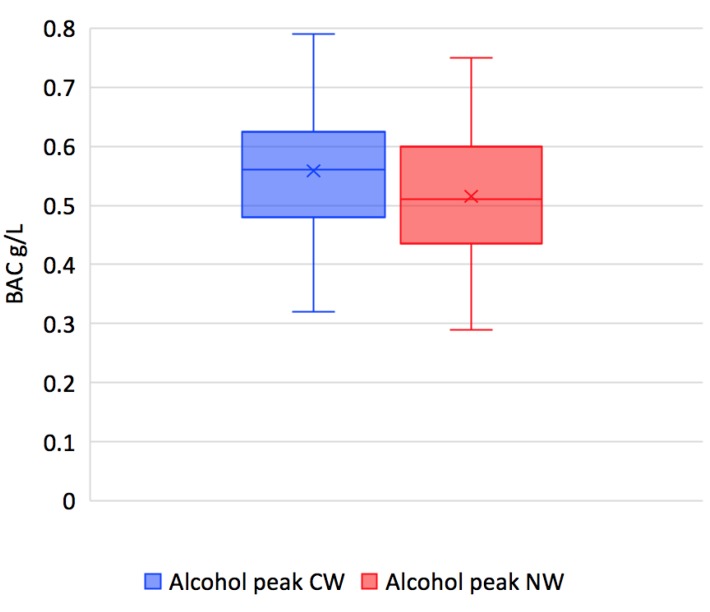
Box-and-whisker diagram of peak BAC levels after the ingestion of the conventional wine (CW) and of the natural wine (NW) (g/L) (*p* < 0.0002).

**Figure 6 nutrients-11-00986-f006:**
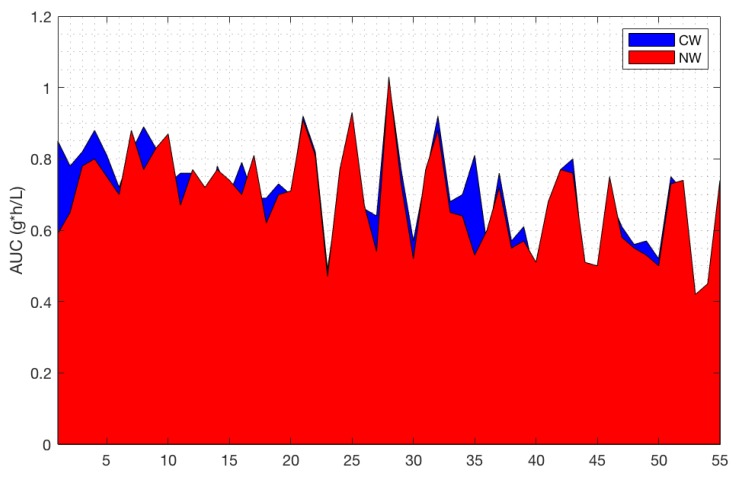
Area under the curve (AUC) calculated for the conventional wine (CW) and the natural wine (NW) (*p* = 0.13).

**Table 1 nutrients-11-00986-t001:** Characteristics of the natural and conventional wines tested.

	Natural Wine	Conventional Wine
Actual alcoholic strength by volume (vol %)	13.2	13
Volatile acidity (mEq/L)	20	4.5
Total sugar content (g/L)	<1.5	<1.5
Total dry extract (g/L)	25	18
Total sulfur dioxide (g/L)	0.02	0.11
Pesticides ^1^	not present	present

^1^ Over 200 pesticides were analyzed. Traces of iprovalicarb (45 µg/kg) and fenhexamid (120 µg/kg) were found in the conventional wine.

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
