# Peer review of "Vive la Difference! The Effects of Natural and Conventional Wines on Blood Alcohol Concentrations: A Randomized, Triple-Blind, Controlled Study"

_nutrients, 2019, doi:10.3390/nu11050986_

Round 1
Reviewer 1 Report
The manuscript Nutrients-462854 describes a study reporting the effect of “natural” and “conventional” wine consumption on blood alcohol content in 55-male individuals. Some differences were found between the two wine effects in terms of resulting blood alcohol level after 20 min. The study is of interest and this type of research often attracts a great media attention. Although it appears to have been carried out with great care, several issues/clarifications should be addressed, at least in my opinion.
The first issue with this study is the poor selection of the experimental wine i.e. Cortese white wine. The authors often (and correctly) remind to the reader the beneficial impact of phenolic substances. In view of this, why did the authors select a white instead of a red wine variety? White grape juice is fermented in absence of skins and seeds. Thus, no much phenolics are expected in the final wine. Cortese is also a very minor white grape, unknown to the most. A much more robust and attracting study was to include a red variety of international interest e.g. Merlot, Shiraz, Cabernet Sauvignon.
The second issue is the low number of wines analysed: 1 each category (2 in total). Indeed, the authors started from a total of 300 wines; but, only 2 wines were chosen at the end. It is not clear why 298 were discarded.
Mostly important, the authors did not provide any information about the wine serving and/or room temperature during drinking. It is very much well known that wine and room temperatures have an impact in the cardiovascular and metabolic responses (e.g. Front. Physiol. 9:1334. doi: 10.3389/fphys.2018.01334). It could be possible that those differences reported by the authors could be due to different wine serving temperatures (?).
Specific comments:
L25-27: how can ~1 g/L of sugar, different yeasts and bacteria impact the blood alcohol content? Any evidence or just speculation?
L2-4: The title is a little bit ambiguous. I am not sure how the exclamation ‘Vive la difference!’ links to the manuscript.
L22: at their peaks…Please, specify or rephrase...unclear (peaks of what?)
L31-65: The authors distinguish natural and conventional wines. This arbitrary definition should be explained somewhere in the introduction (but keep it in L68-71).
Table 1. L77: Traces?? Specify, the limit of detection of the method (e.g.<1 ng/L).
L71: 300 wines were purchased and only 2 were found suitable for testing. What criteria the authors adopted to discard 298 wines?
L83: ‘…any process to add or subtract ingredients…’ incorrect. The correct term is ‘fining’
L85: Please, change ‘farmed’ to ‘grown’
L88: traces? Specify the LOD
The Section ‘2.2. Study design’ is uncomplete. There is not information in the manuscript how randomization occurred. This is the paragraph where all this information and detail should be provided about this ‘randomized’ design. Please, move L122-126 to Section 2.2.
L121: how was this code number (3 digit?) assigned and generated?
L142-143: Please, move this ethical approval to L95.
Figure 1: L154: Examples of blood alcohol concentration.
Figures 2-5: y axis label is missing. Please, add it in each figure.
L171-173: what figure the authors are discussing here?? It is not specified.
L194: Please, add a reference
L202 & 162: significantly? Please, clarify (or rephrase) as significantly imply some sort of statistical meaning.
Author Response
Reviewer #1
General Comments from Reviewer
The manuscript Nutrients-462854 describes a study reporting the effect of “natural” and “conventional” wine consumption on blood alcohol content in 55-male individuals. Some differences were found between the two wine effects in terms of resulting blood alcohol level after 20 min. The study is of interest and this type of research often attracts a great media attention. Although it appears to have been carried out with great care, several issues/clarifications should be addressed, at least in my opinion.
The first issue with this study is the poor selection of the experimental wine i.e. Cortese white wine. The authors often (and correctly) remind to the reader the beneficial impact of phenolic substances. In view of this, why did the authors select a white instead of a red wine variety? White grape juice is fermented in absence of skins and seeds. Thus, no much phenolics are expected in the final wine. Cortese is also a very minor white grape, unknown to the most. A much more robust and attracting study was to include a red variety of international interest e.g. Merlot, Shiraz, Cabernet Sauvignon.
The second issue is the low number of wines analysed: 1 each category (2 in total). Indeed, the authors started from a total of 300 wines; but, only 2 wines were chosen at the end. It is not clear why 298 were discarded.
Mostly important, the authors did not provide any information about the wine serving and/or room temperature during drinking. It is very much well known that wine and room temperatures have an impact in the cardiovascular and metabolic responses (e.g. Front. Physiol. 9:1334. doi: 10.3389/fphys.2018.01334). It could be possible that those differences reported by the authors could be due to different wine serving temperatures (?).
Replies to the general comments from the Reviewer
We would like to thank the Reviewer for the positive evaluation of our work and for the many useful comments on how to improve the manuscript.
The first issues raised by the Reviewer were the selection of the experimental wine and the number of wines analysed. Our decision to select these two Cortese white wines was both reasoned and deliberate: i) we wanted to conduct the study using wines in which the effect of polyphenolic content would be minimized and differences due to sulfitation could emerge, despite the small quantities of wine administered, and white wines were the logical candidates. ii) we wanted to be able to directly verify the methods of cultivation and winemaking adopted by the two different wine producers, and therefore decided to screen varieties grown in the region where the study was conducted (Piedmont), where international varieties are rare and above all are discouraged in favour of traditional local grapes; iii) the selection of the two wines started with a short list of 50 natural wines and 250 conventional wines produced in Piedmont. Our aim was to find a pair of wines, one natural and one conventional, with the best-possible correspondence in grape variety, proximity of area of production, age of harvesting, alcoholic strength by volume, and both having a low sugar content (<1.5 g/L). Unlike the other wines screened, the two Cortese white wines ultimately selected for the study fully satisfied the inclusion criteria, and while not as familiar to most, were extremely well-suited for the purposes of our study.
The other main issue raised by the Reviewer concerned the wine serving and room temperature during drinking. We are grateful to the Reviewer for having pointed out this lacuna in the text. In fact, not only were the two samples both served at 21°C, but the temperature of the room was also maintained at a constant 21°C throughout the entire testing period. The manuscript has been updated accordingly and the paper by Sarafian et al., 2018, suggested by the Reviewer, is now included in the References.
A detailed point-to-point reply to the Reviewer's comments is provided below.
Replies to the specific comments from the Reviewer
L25-27: how can ~1 g/L of sugar, different yeasts and bacteria impact the blood alcohol content? Any evidence or just speculation?
Our Reply:Thank you for having pointed out the mistake in the Abstract: as previously mentioned, the sugar content was low in both samples, so significant differences due to sugar content were not expected, and the wording in the Abstract has been corrected to reflect this. As for the impact of different yeasts and bacteria, this topic is addressed in section #4. A reference has been added to the discussion, reported below, in order to corroborate this hypothesis (Romboli, Y.; Mangani, S.; Buscioni, G.; Granchi, L.; Vincenzini, M. Effect of Saccharomyces cerevisiae and Candida Zemplinina on quercetin, vitisin A and hydroxityrosol contents in Sangiovese wines. World J Microbiol Biotechnol 2015, 31:10137-45).
Another important distinction between natural and conventional wines lies in the vinification process. Natural wine is the product of spontaneous fermentation by indigenous yeasts naturally found on the grapes, while conventional wines are produced using mixtures of laboratory-selected microorganisms. The presence of various strains of bacteria and yeasts during fermentation results in the development of different metabolites [20]. At present, the results of chemico-physical analysis of the samples used in this study are unable to provide precise information about these differences. Additional data may emerge thanks to the use of new technologies. In recent years, for example, high-field 1H nuclear magnetic resonance (1H-NMR) spectroscopy has allowed detailed investigation of wine metabonomics [21] and has demonstrated that the vinification protocol is one of the chief factors determining the amino acid, alcohol and polyphenol make-up of two wines from the same geographic location [22]. Another study established that different production chains determine variations in the amount and type of antioxidants found in organic and biodynamic wines [23]. Comparable data on wines produced using the natural winemaking process are not yet available.
Besides producing wines with different amino acid and polyphenolic profiles, differences in the natural and conventional fermentation pathways may also generate other molecules that interact with absorption or with specific isoforms of alcohol dehydrogenase (ADH), the enzyme involved in breaking down alcohol. This would in turn lead to differences in the rates of metabolism of natural wine and conventional wine.
L2-4: The title is a little bit ambiguous. I am not sure how the exclamation ‘Vive la difference!’ links to the manuscript.
Our Reply:The use of this familiar expression evokes France’s world-renowned culture of wine and gastronomy and intends to underline that the study detected differences, shown at T20 and at their maximum Blood Alcohol Concentration (peak), between two wines obtained through different agronomical and productive techniques, also serving as a subtle exhortation of the importance of agricultural and nutritional biodiversity.
L22: at their peaks…Please, specify or rephrase...unclear (peaks of what?)
Our Reply:The sentence has been rephrased using the expression “maximum concentration” to refer to the highest value of Blood Alcohol Concentration detected (see Fig.2).
L31-65: The authors distinguish natural and conventional wines. This arbitrary definition should be explained somewhere in the introduction (but keep it in L68-71).
Our Reply:We thank the Reviewer for pointing this out and have modified the manuscript accordingly.
Table 1. L77: Traces?? Specify, the limit of detection of the method (e.g.<1 ng/L).
Our Reply:We are grateful to the Reviewer for pointing this out. The values of the pesticides detected have been included in the table.
L71: 300 wines were purchased and only 2 were found suitable for testing. What criteria the authors adopted to discard 298 wines?
Our Reply:Please see our reply to the Reviewer’s comments above for an explanation of this issue and a description of the changes made to the manuscript.
L83: ‘…any process to add or subtract ingredients…’ incorrect. The correct term is ‘fining’
Our Reply:The term has been corrected.
L85: Please, change ‘farmed’ to ‘grown’
Our Reply:The word “farmed” has been changed to “grown”.
L88: traces? Specify the LOD
Our Reply:The LOD has been specified.
The Section ‘2.2. Study design’ is uncomplete. There is not information in the manuscript how randomization occurred. This is the paragraph where all this information and detail should be provided about this ‘randomized’ design. Please, move L122-126 to Section 2.2.
Our Reply:We have moved the lines mentioned to Section 2.2 and added the following details about how randomization occurred: Among the 167 eligible subjects, 55 subjects were drawn by a random number generator. In addition, as the test was starting,the matricescontaining the matching between the subjects and the bottles were generatedby a random number generator, as well asthe assignment of one of the two types of wine to each subject in the first or second week.
L121: how was this code number (3 digit?) assigned and generated?
Our Reply:The following explanation has been added to the text: The labels on the bottles were masked, and the wines were identified strictly by a 4-digit code number. The first two digits indicated the day on which the test was performed, while the second two indicated the bottle index randomly assigned to each bottle.
L142-143: Please, move this ethical approval to L95.
Our Reply:The ethical approval has been moved as recommended.
Figure 1: L154: Examples of blood alcohol concentration.
Our Reply:The legend has been modified according to the Reviewer’s suggestion.
Figures 2-5: y axis label is missing. Please, add it in each figure.
Our Reply:The missing y axis labels have been added to each figure.
L171-173: what figure the authors are discussing here?? It is not specified.
Our Reply:We thank the Reviewer for noting this oversight. The reference to Appendix A has been added.
L194: Please, add a reference
Our Reply:The reference has been added.
L202 & 162: significantly? Please, clarify (or rephrase) as significantly imply some sort of statistical meaning.
Our Reply:In L162 “significantly” has been maintained because it is used in the statistical sense (as the p value is<0.05, and more precisely, 0.012). In L202 the word “significant” has been replaced by the word “substantial”.
Reviewer 2 Report
Dear authors:
Thank you for the opportunity to review your manuscript. I found it of great interest; however some methodological issues must be addressed.
Study design: The study design is not explained in detail.
1.- RANDOMIZATION: How did randomization take place? Participants in the study were randomly assigned to what? Was the allocation concealed?
2.- TRIAL DESIGN: Please further explain the design of the trial. Is it a crossover trial? Discuss the necessity of a washout period. Is there any difference between both periods that may have affected the results?
3.- TRIPLE-BLIND: triple-blind is usually used to describe trials in which the participant, the individual who administered the intervention and the individual who assess the outcome are blind to the intervention.
4.- INFORMED CONSENT: It seems not compatible that participants provided informed consent and that they were “unaware that the wine they were being given was different from that of the previous week”.
5.- VALIDITY: A questionnaire was used to measure height, weight, BMI, dietary habits… Please discuss the validity of this questionnaire.
6.- CHARACTERISTICS OF PARTICIPANTS: Please describe characteristics of participants besides age, weight, height and BMI. These results may not be extrapolated to other populations with different characteristics.
7.- REQUIREMENTS: Participants were required to abstain from drinking for 7 days… Was this also true for the second wine consumption? Did author check if participants actually complied with the requirements?
8.- SUGAR CONTENT: line 80: I don’t see why sugar content is similar for both wines:<1.0 may be 0.99 or 0.
9.- TIME: At the experiment, time started when the participant finished the glass of wine? Please specify. Was there any difference between time needed to consume the glass of natural wine and the glass of conventional wine?
10.- STATISTICS: The sample size calculation is poorly explained. When authors say “to obtain a 90% confidence level” they mean a 10% alpha error or a 90% power (10% beta error). The study design requires a paired hypothesis testing. Please use paired Student t test instead of independent Student t test. Line 145-147, use p25 and p75 instead of Q1 and Q3.
11.- FIGURE 1: Why did authors described these 4 participants? Is the decision based in any criteria? Please explain.
12.- FIGURE 2: I would recommend to use the same scale as in the other plots (0-0.8). Please add axis titles.
13.- Line 163: data in the text do not fit data in the graph. Please check the discordance.
14.- Is there an explanation to the higher variability showed for CW vs NW in figure 3?
15.- Line 178: “This parameter proved not to be significant” In which terms? If you mean statistically significant, please provide a p-value.
16.- Figure 5: Please further explain what is plot in this figure. Also add axis titles. Please review p<0.19 (do authors mean p=0.19?). What test was used to obtain p=0.19? Please describe in the methods section.
17.- CONCLUSIONS: To conclude that there will be differences in terms of risk of intoxication, or risk of car accident is overdimensioned. Please tone down the implications of the results.
18.- Line 268: include the reference to Bassani et al.
19.- Line 274: authors did not compare metabolism. Authors studied the pharmacokinetics of alcohol after consumption of two different wines.
20.- Line 63: 24 g of alcohol
Author Response
Reviewer #2
General Comments from Reviewer
Thank you for the opportunity to review your manuscript. I found it of great interest; however some methodological issues must be addressed.
Study design: The study design is not explained in detail.
Replies to the general comments from the Reviewer
We would like to thank the Reviewer for the positive evaluation of our work and for the many useful comments on how to improve the manuscript. A detailed point-to-point reply to the reviewer's comments is provided below.
Replies to the specific comments from the Reviewer
1.- RANDOMIZATION: How did randomization take place? Participants in the study were randomly assigned to what? Was the allocation concealed?
Our Reply:The following explanation has been added to the text: The study was a randomized, triple-blind, controlled trial.Each phase of the trial was conducted in triple blind, meaning that at no time were any of the three research teams aware of the identity of the wines being administered to the subjects or being analyzed. The first team designed the study, selected the volunteers and set up the samples for testing. The second administered the doses of wine to the subjects and recorded the resulting data, and the third conducted the final data analysis. Among the 167 eligible subjects, 55 subjects were drawn by a random number generator. In addition, as the test was starting,the matrices containing the matching between the subjects and the bottles were generatedby a random number generator, as well as the assignment of one of the two types of wine to each subject in the first or second week.
2.- TRIAL DESIGN: Please further explain the design of the trial. Is it a crossover trial? Discuss the necessity of a washout period. Is there any difference between both periods that may have affected the results?
Our Reply:Please see the above reply in response to specific comment #1. Section 2.2 of the manuscript has been expanded in order to explain the design of the trial. This was a crossover trial. The washout period was necessary for minimizing the possible interference of assuming other doses of alcohol during the previous week. The washout period, also in the week before the test, was exactly designed to avoid any possible metabolic difference in the two periods preceding the test.
3.- TRIPLE-BLIND: triple-blind is usually used to describe trials in which the participant, the individual who administered the intervention and the individual who assess the outcome are blind to the intervention.
Our Reply:Yes, it was a triple-blind test. This has now been clarified in section 2.2 (study design).
4.- INFORMED CONSENT: It seems not compatible that participants provided informed consent and that they were “unaware that the wine they were being given was different from that of the previous week”.
Our Reply:The participants signed consent to assume a maximum of 2 Units of alcohol per day of test of a wine authorized for commerce in Italy, without indicating the specific type of wine. The wording of section 2.4 has been modified to avoid suggesting incompatibility with informed consent.
5.- VALIDITY: A questionnaire was used to measure height, weight, BMI, dietary habits… Please discuss the validity of this questionnaire.
Our Reply:The participants double signed under their own responsibility the truthfulness of the data they declared.
6.- CHARACTERISTICS OF PARTICIPANTS: Please describe characteristics of participants besides age, weight, height and BMI. These results may not be extrapolated to other populations with different characteristics.
Our Reply:The sample is described in paragraph 3 The 55 male subjects recruited for the study were of median age 23 (p25: 21 years, p75: 24 years), median weight 69 kg (p25: 65 kg, p75: 78 kg), median height 178 cm (p25: 174 cm, p75: 183 cm), and median BMI 22 kg/m2(p25: 20.8 kg/m2cm, p75: 22.9 kg/m2). We selected a homogeneous sample of young healthy males to minimize confounding factors related to sex, physiological differences, individual kinetics, morbidities, or drug therapies. The Conclusion indicates the need for further investigations (see par. 5).
7.- REQUIREMENTS: Participants were required to abstain from drinking for 7 days… Was this also true for the second wine consumption? Did author check if participants actually complied with the requirements?
Our Reply:The same requirements was asked also for the second week, when the test was performed at day +7 from the first test. The participants double signed under their responsibility the truthfulness of the data declared.
8.- SUGAR CONTENT: line 80: I don’t see why sugar content is similar for both wines:<1.0 may be 0.99 or 0.
Our Reply:We definitely thank with Reviewer has the text din’t sound clear. The study was intended to test two wines with a low sugar content (<1.5 g/L), in order to minimize the possible interference of sugar content on the alcohol kinetic. The manuscript was changed accordingly with the Reviewer’s comment.
9.- TIME: At the experiment, time started when the participant finished the glass of wine? Please specify. Was there any difference between time needed to consume the glass of natural wine and the glass of conventional wine?
Our Reply:For both wines, the subjects had three minutes within which to drink the wine; the test time started at the end of this 3 minutes period.
10.- STATISTICS: The sample size calculation is poorly explained. When authors say “to obtain a 90% confidence level” they mean a 10% alpha error or a 90% power (10% beta error). The study design requires a paired hypothesis testing. Please use paired Student t test instead of independent Student t test. Line 145-147, use p25 and p75 instead of Q1 and Q3.
Our Reply:We are grateful to the Reviewer for this extremely useful suggestion. The manuscript has been modified accordingly.
11.- FIGURE 1: Why did authors described these 4 participants? Is the decision based in any criteria? Please explain.
Our Reply:The curves were selected randomly. In order to comply with the Reviewer’s comment, we’ve used a random number generator to select 4 new participants to replace the 4 participants previously cited in the manuscript.
12.- FIGURE 2: I would recommend to use the same scale as in the other plots (0-0.8). Please add axis titles.
Our Reply:Fig.2 has been corrected as suggested by the Reviewer.
13.- Line 163: data in the text do not fit data in the graph. Please check the discordance.
Our Reply:The data in the text has been corrected and now fit the data in the graph.
14.- Is there an explanation to the higher variability showed for CW vs NW in figure 3?
Our Reply:We very much appreciate the Reviewer’s question, which allowed us to discover an error in Figure 3. The image has been corrected and updated in the manuscript. Further investigations are needed to better explain differences in metabolism of natural and conventional wines.
15.- Line 178: “This parameter proved not to be significant” In which terms? If you mean statistically significant, please provide a p-value.
Our Reply:The p-value has been added in the manuscript.
16.- Figure 5: Please further explain what is plot in this figure. Also add axis titles. Please review p<0.19 (do authors mean p=0.19?). What test was used to obtain p=0.19? Please describe in the methods section.
Our Reply:The figures have been corrected and completed according to the Reviewer’s comment. The test used was Student’s t-test for paired data, as now cited in the manuscript.
17.- CONCLUSIONS: To conclude that there will be differences in terms of risk of intoxication, or risk of car accident is overdimensioned. Please tone down the implications of the results.
Our Reply:The text has been modified accordingly and the implications of the results toned down.
18.- Line 268: include the reference to Bassani et al.
Our Reply:The study by Bassani is under preparation. We referred to this study in the manuscript as a “personal communication”.
19.- Line 274: authors did not compare metabolism. Authors studied the pharmacokinetics of alcohol after consumption of two different wines.
Our Reply:The original text was inaccurate, and “metabolism” has been changed to “pharmokinetics”.
20.- Line 63: 24 g of alcohol
Our Reply:The manuscript has been corrected.
Round 2
Reviewer 1 Report
The authors have adequately addressed the comments of this reviewer and the manuscript is suitable for publication (to me at least). A few minor changes still to be made
L60: at 20 and 40 mins
L114-115: see Materials and Methods)
L158: with near-perfect (or the best possible)
L189: ..., 55 subjects were drawn ...
L174: fining agents processes
L234: 21 ± 1°C
L237: the two tasting sessions...
L284: ..., in subjects #21, etc..
Table 1: the level of precision (i.e. decimals) of those values can be adjusted to 13.2 & 13 (for alcohols); 25 & 18 (total dry extract); 0.02 (TSO2)..
L458: ...both from the same producing area and variety, similar alcohol content and residual sugar
L459: 20 and 40 mins
Author Response
We thank the auditor for all the comments, which were fully taken on board. The manuscript has been modified accordingly.
Reviewer 2 Report
Authors did not sufficiently answer to questions 1, 2, 5, 6 and 8 of my previous report.
Please, explain the experiment so any reader could replicate it.
Author Response
REV #2
General comments from Reviewer:
Authors did not sufficiently answer to questions 1, 2, 5, 6 and 8 of my previous report.
Please, explain the experiment so any reader could replicate it.
Our Reply:
We thank the Reviewer for the opportunity of making the manuscript cleare. Our aim in this second round of revisions, was to make the experiment clearly replicable to any reader. We’ve explained and detailed better any methodological issue, as detailed below in the point-to-point replies. The manuscript was modified accordingly.
1.- RANDOMIZATION: How did randomization take place? Participants in the study were randomly assigned to what? Was the allocation concealed?
Our Reply: The Study Design section has been reformulated in the manuscript to make it clearer, as follows:
2.2. Study design
The study was a randomized, triple-blind, controlled trial. Each phase of the trial was conducted in triple-blind, as the type of wine administerd was unknown to (a) the research participants, (b) the individuals who administer the wines, and (c) the individuals who assess the outcomes.
Three teams worked on the experiment. The first team designed the study, selected the volunteers and set up the samples for testing. Any label on the bottles of wines used for the experiment was masked and the wines were identified strictly by a 4-digit code number. The first two digits indicated the day on which the test was performed, while the second two indicated the bottle index randomly assigned to each bottle. The second team administered the doses of wine to the subjects and recorded the resulting data, and the third team processed the data, cross-referencing the matrix of wine types received from the first team with the BAC data of the individual participants, collected by the second team. The second team wasn’t aware of the type of the wine that administered to the participants and the third team never met the participants.
As in a crossover trial, each participants serving as their own control, the participants intended to receive one type of wine or the other the first week (and the other wine the following week) were chosen randomly. Among the 167 eligible subjects, 55 subjects were drawn by a random number generator. In addition, at the start of the test, the matrices containing the matching between the subjects and the bottles were generated by a random number generator. The researchers in the first team, who were in charge of selecting the subjects, did not know which wine the subjects would be assigned to, because the matrix with the assignments was in the hands of the second team. The subjects were also unaware of the type of wine they would drink, because they were informed that they would have to drink a sample of wine every week, without specifying either the type or whether it was the same or different.
2.- TRIAL DESIGN: Please further explain the design of the trial. Is it a crossover trial? Discuss the necessity of a washout period. Is there any difference between both periods that may have affected the results?
Our Reply:The Study Design section has been reformulated in the manuscript to make it clearer. As now better explained in section 2.2 each subjects served has thier own control, assuming the “gold standard”, conventional wine, one week, and the “novel-product”, natural wine, the other week.
The washout period was necessary for minimizing the possible interference of assuming other doses of alcohol during the previous week. The washout period is needed to minimize the possibile interference of increased metabolism of alcohol in cronic consumer, that is known in alcoholic subjects but suspected also, at lower rates, in regular consumers, [as described in his Review “Alcohol Metabolism” the by Arthur I Cederbaum, (Clin Liver Dis. 2012 Nov; 16(4): 667–685), cited in ref #13 of our manuscript]. One week was assumed as an acceptable washout period for a moderate drinker (https://www.epicentro.iss.it/passi/indicatori/alcol - The epidemiology portal for public health by the Italian National High Institute of Public Health.
5.- VALIDITY: A questionnaire was used to measure height, weight, BMI, dietary habits… Please discuss the validity of this questionnaire.
Our Reply: We used a questionnaire (PASSI) validated and promoted by the Italian Ministry of Health, the National High Institute of Public Health and National Centre for Disease Prevention and Control for the surveillance of behavioural risk factors and for the monitoring of chronic disease prevention programmes. The questionnaire is divided into sections and only those relating to the topic under investigation (anthropometry, general health and alcohol consumption) have been proposed. The data collected with the PASSI questionnaire are the basis for national and regional reports on alcohol consumption in Italy, where the study was carried out.
The PASSI questionnaire can be downloaded at the following link and was cited in the updated manuscript.
6.- CHARACTERISTICS OF PARTICIPANTS: Please describe characteristics of participants besides age, weight, height and BMI. These results may not be extrapolated to other populations with different characteristics.
Our Reply: The manuscript was updated:The sample consisted of healthy Caucasian Males aged 18 to 30, enrolled at the Polytechnic of Turin,with a BMI between 18.5 kg/m2and 25 kg/m2,who did not take any drug therapy and, who consumed an average of 4 alcohol units per week, who could understand the purpose of the study and thus provide written informed consent. Exclusion criteria:use of prescription medicine for chronic conditions; any pathology that might interfere with alcohol metabolism; habitual consumption of more than 4 units of alcohol per week. The 55 male randomly selected out of the 167 elegible subjects recruited, were of median age 23 (p25: 21 years, p75: 24 years), median weight 69 kg (p25: 65 kg, p75: 78 kg), median height 178 cm (p25: 174 cm, p75: 183 cm), and median BMI 22 kg/m2(p25: 20.8 kg/m2cm, p75: 22.9 kg/m2).
The PASSI questionnaire is unanimously recognized in Italy as valid for extrapolating behavioral indications and health prescriptions to the general population.
8.- SUGAR CONTENT: line 80: I don’t see why sugar content is similar for both wines:<1.0 may be 0.99 or 0.
Our Reply:The legal limit for defining the sugar content of a wine in Italy is set at 3g/L and laboratories do not conventionally provide measures below 1g/L. For the 248ml we administer, the difference of 1 gram per litre is equivalent to 0.25g of sugar, values that in no way interfere with the parameters we evaluate. However, we have accepted the suggestion of the Reviewer and any reference to “similar sugar content” has been deleted from the text and replaced by “low sugar content”.